# Genome-Wide Identification of Rubber Tree *SCAMP* Genes and Functional Characterization of *HbSCAMP3*

**DOI:** 10.3390/plants13192729

**Published:** 2024-09-29

**Authors:** Baoyi Yang, Xiao Huang, Yuanyuan Zhang, Xinsheng Gao, Shitao Ding, Juncang Qi, Xiangjun Wang

**Affiliations:** 1The Key Laboratory of Oasis Eco-Agriculture, Xinjiang Production and Construction Group, Agricultural College, Shihezi University, Shihezi 832003, China; 20192312016@stu.shzu.edu.cn; 2Rubber Research Institute, Chinese Academy of Tropical Agricultural Sciences, Haikou 571101, China; 13976786119@163.com (X.H.); zhangyuanyuan@catas.cn (Y.Z.); gaoxinsheng@catas.cn (X.G.); 22220951310004@hainanu.edu.cn (S.D.)

**Keywords:** rubber tree, *secretory carrier membrane protein* (*SCAMP*) gene family, stress response, *HbSCAMP3*, plant height regulation

## Abstract

Natural rubber produced by the rubber tree is a vital industrial raw material globally. Seven *SCAMP* gene family members were identified in the rubber tree, and the phylogenetic tree classified *HbSCAMPs* into three subfamilies. Significant differences were observed among *HbSCAMPs* in terms of gene length, number of exons, and composition of conserved motifs. The expansion of *HbSCAMPs* in the rubber tree genome is associated with segmental duplications. The high expression of *HbSCAMP1–6* in petioles and *HbSCAMP7* in stem tips, along with their distinct responses to drought, salt, and wound stresses, indicates their crucial roles in substance transport and stress adaptation. Transgenic poplar experiments demonstrated that overexpression of *HbSCAMP3* significantly promotes plant height growth, with localization in the tobacco plasma membrane, suggesting its involvement in regulating plant growth through membrane transport processes. These findings enhance the understanding of *HbSCAMPs* in rubber trees and provide new insights into how plants finely tune gene family members to adapt to environmental changes.

## 1. Introduction

The rubber tree (*Hevea brasiliensis*) is a critical source of natural rubber on a global scale, with approximately 90% of the world’s natural rubber production derived from it [1]. However, abiotic stresses, including drought, low temperature, salinity, and mechanical damage, pose a significant threat to the expansion of rubber trees into non-traditional areas [2]. These stresses impact the physiological and biochemical processes of the plant, triggering the expression of stress response genes, such as *superoxide dismutase (SOD)* genes, which play a crucial role in protecting plant cells from oxidative damage caused by reactive oxygen species [3]. Furthermore, the identification and characterization of protein kinases (PK) in rubber trees have demonstrated their involvement in stress response mechanisms, emphasizing the significance of cellular regulation under adverse conditions to prevent a considerable reduction in rubber yield [4]. Consequently, comprehensive research into the mechanisms by which rubber trees respond to abiotic stresses is of paramount importance for enhancing their resistance and resilience.

The growth of plant height is also a significant factor that affects the cultivation and production of rubber trees. The height growth of rubber trees is not only associated with the yield of natural rubber but also affects the utilization value of rubber wood. Effective height control is of paramount importance for natural rubber production. Growth regulators such as paclobutrazol not only reduce plant height but also enhance desirable traits such as stem girth, bark thickness, and latex vessel development, thereby increasing the yield of natural rubber [5]. In the context of timber production, the management of tree height is of equal importance, as it influences the formation of reaction wood and, consequently, the biosynthesis of lignin—a key secondary metabolite in the phenylpropanoid pathway, playing a pivotal role in plant structure and function [6]. Recent advances in molecular biology have elucidated the regulatory roles of differentially expressed genes (DEGs) and transcription factors (TFs), including *MYB*, *C2H2*, and *NAC*, in controlling lignin biosynthesis in rubber trees [7]. Despite progress in understanding growth regulation mechanisms through approaches such as genome-wide association studies (GWAS) and high-resolution mapping of quantitative trait loci (QTLs) [8,9], a comprehensive understanding of the factors governing the height of rubber trees remains incomplete.

*Secretory carrier membrane proteins (SCAMPs)* are of great importance in the physiological processes of plant cells, particularly in response to abiotic stresses. It has been demonstrated that SCAMP proteins play a role in the regulation of membrane and vesicular transport, which is essential for the signal transduction of stress responses in plants [10]. The detrimental effects of salt stress on plants are primarily attributed to the toxic effects of Na^+^ [11], and SCAMP proteins play a role in the transport of Na^+^ ions, particularly during vesicular transport processes [12]. In soybean, under salt stress conditions, leaves of GmSCAMP5-OE plants exhibited less wilting, less Trypan blue staining, and significantly lower Na^+^ content compared to the EV-Control plants. Conversely, GmSCAMP5-RNAi plants displayed more Trypan blue staining and significantly higher Na^+^ content in the leaves than the EV-Control plants [13]. Apart from salt stress, there are currently no reports on *SCAMP* response to other abiotic stresses. SCAMP proteins also play an important role in the regulation of plant growth and development. In plants, *SCAMPs*, as integral membrane proteins located at the plasma membrane (PM) and the trans-Golgi network (TGN)/early endosomes, are involved in endocytic transport processes, which are vital for plant cells to adapt to environmental changes and maintain intracellular homeostasis [14]. In poplar trees, SCAMP proteins have been demonstrated to influence the accumulation of secondary cell wall components, including polysaccharides and phenolic compounds. This suggests that *SCAMPs* play a role in regulating the abundance of cell wall precursors and participate in the transport of proteins involved in cell wall biosynthesis through the regulation of membrane transport mechanisms [15]. It is therefore of significant importance to conduct a comprehensive investigation into the functions of *SCAMPs* in plants in order to gain insight into the regulatory mechanisms of plant growth and development, as well as the mechanisms by which plants resist abiotic stresses.

This study employed a whole-genome analysis of the rubber tree genome, utilizing SCAMP protein sequences from *Manihot esculenta*, *Populus trichocarpa*, and *Arabidopsis thaliana*. Seven putative *HbSCAMP* sequences were detected in the rubber tree genome, named *HbSCAMP1* to *HbSCAMP7*. Their gene structures, chromosomal locations, and phylogenetic relationships were analyzed. qRT-PCR was used to examine the tissue-specific expression and responses of *HbSCAMPs* under stress conditions. An overexpression vector for *HbSCAMP3* was constructed and introduced into poplar to explore its role in plant growth. This study is the first comprehensive genomic analysis of *HbSCAMPs*. These results provide valuable information on *HbSCAMPs* in rubber trees and offer potential candidate genes and a research foundation for targeted genetic breeding in plants.

## 2. Results

### 2.1. Whole-Genome Characterization of SCAMP Proteins in Rubber Trees

To delineate the attributes of the SCAMP protein family in the rubber tree, a Hidden Markov Model (HMM) focused on the conserved SCAMP domain was employed to conduct an exhaustive search of the rubber tree proteome. This approach resulted in the identification and validation of seven unique SCAMP proteins, named HbSCAMP1 to HbSCAMP7, according to their genomic sequencing (Figure 1 and Appendix A).

The HbSCAMP proteins were further analyzed for key physicochemical properties, including amino acid (aa) sequence length, molecular weight (MW), isoelectric point (pI), instability index, and GRAVY score. The protein sequence length ranged from 193 in HbSCAMP3 to 462 in HbSCAMP2, averaging 302.1 aa across the family. Corresponding MW values ranged from 21.3 kDa for HbSCAMP3 to 52.7 kDa for HbSCAMP2, with an average of 34.2 kDa, highlighting HbSCAMP2 as the largest member in terms of both aa count and MW.

Furthermore, the pI values of the HbSCAMP proteins ranged from 5.8 to 6.73, indicating a predominantly acidic to neutral isoelectric character. The instability index, which measures protein stability, ranged from 42.88 to 50.94, suggesting moderate stability across the proteins. The GRAVY scores, which indicate overall hydrophobicity, varied from −0.033 to 0.443, revealing a spectrum of hydrophilic to hydrophobic characteristics.

### 2.2. Phylogenetic Relationship, Gene Structure, and Conserved Motif Analysis of HbSCAMPs

To trace the evolutionary relationships within the *SCAMP* gene family across species, an ML phylogenetic tree was constructed to elucidate their diversification and functional divergence (Figure 2). The tree organized 26 *SCAMP* genes from *H. brasiliensis*, *A. thaliana*, *M. esculenta*, and *P. trichocarpa* into three subfamilies, with a balanced distribution of *H. brasiliensis* genes suggesting a conserved role for *SCAMP* genes across species (Appendix A).

The gene structure and conserved motifs of the *SCAMP* family in the rubber tree were analyzed to reveal patterns linked to phylogenetic relationships (Figure 3a–d). The MEME tool identified 15 conserved motifs with high similarity within subfamily members, suggesting potential shared functions. All HbSCAMP proteins contained the SCAMP domain, with HbSCAMP2 also containing the DUF707 domain. Exon/intron structure analysis revealed that the *HbSCAMP* genes varied in exon number from 1 to 14, with subfamily III members having 13 exons, subfamily I members having 12 and 11 exons, and subfamily II members having 14 and 1 exon (Figure 3d). This diversity may relate to gene complexity and functional differences.

The promoter regions of the *HbSCAMP* genes were analyzed for cis-regulatory elements (CREs) using PlantCARE, identifying 1356 CRE sites, of which 170 were classified (Figure 4a, Appendix A). The family was mainly associated with hormone response, growth and development, and stress response elements, with light-responsive elements being the most abundant. *HbSCAMP6* had the highest number of CREs at 32, while *HbSCAMP1* had the fewest, at 13, indicating complex regulatory networks for the *HbSCAMP* genes.

### 2.3. Evolutionary Analyses of the HbSCAMPs within and between Species

In order to explore the evolutionary forces that have influenced the *SCAMP* gene family and their role in speciation, a detailed evolutionary study was conducted, focusing on the detection of gene duplication events within the *HbSCAMP* genes of the rubber tree. Such events are pivotal for elucidating the processes of gene diversification and functional specialization (Figure 5). Using the MCScanX approach, we discovered four pairs of segmentally duplicated genes within the *HbSCAMP* family, highlighting the role of duplication in the evolutionary history of the family. The Ka/Ks ratios for these duplicated pairs were all less than one (Appendix A), indicating that they are subject to purifying selection, which suggests evolutionary constraints to preserve their biological functions.

In addition, comparative evolutionary analyses with *A. thaliana*, *M. esculenta*, and *P. trichocarpa* shed light on the timing and mechanisms of gene duplication in the *HbSCAMP* lineage (Figure 5). The findings indicated a closer evolutionary relationship between the rubber tree and other woody species such as *M. esculenta* and *P. trichocarpa*, as opposed to the herbaceous *A. thaliana*. Specifically, five *HbSCAMP* genes showed sequence homology with genes from *M. esculenta* and *P. trichocarpa*, while three *HbSCAMP* genes displayed homology with genes in the *A. thaliana* genome. Importantly, three *HbSCAMP* genes (*HbSCAMP1*, *HbSCAMP3*, *HbSCAMP7*) showed strong evolutionary relationships across the three species examined, highlighting their potential as important evolutionary markers within the *SCAMP* gene family.

### 2.4. Expression Profiles of HbSCAMPs under Stress

In this study, we used qRT-PCR technology to analyze the expression patterns of seven members of the *HbSCAMP* gene family in different tissues of the rubber tree and to further explore their responses to drought, salt, and wound stress conditions (Figure 6). The members of the *HbSCAMP* gene family showed different expression patterns in different tissues of the rubber tree. *HbSCAMP1*, *HbSCAMP2*, *HbSCAMP3*, *HbSCAMP4*, *HbSCAMP5*, and *HbSCAMP6* were highly expressed in petioles, whereas *HbSCAMP7* was highly expressed in stem tips. The expression patterns of the *HbSCAMP* gene family members under drought, salt, and wound stresses exhibited complex changes. In stems, all genes were significantly downregulated following each stress treatment. In petioles, *HbSCAMP1*–*6* were significantly downregulated after each stress treatment, while *HbSCAMP7* showed significant upregulation. Under drought stress, *HbSCAMP1*, *HbSCAMP2*, *HbSCAMP3*, *HbSCAMP6*, and *HbSCAMP7* were significantly upregulated in leaves. Under salt stress, *HbSCAMP1*–*6* were significantly downregulated in leaves, petioles, and stems, while *HbSCAMP7* exhibited the opposite pattern. Under wound stress, the expression of all genes was downregulated in stem tips, leaves, petioles, and stems, but *HbSCAMP1*, *HbSCAMP2*, *HbSCAMP5*, and *HbSCAMP7* were upregulated in bark, whereas *HbSCAMP3* and *HbSCAMP4* were downregulated in bark.

### 2.5. Overexpressing HbSCAMP3 Increased Plant Height in Poplar

To further understand the role of *HbSCAMP* genes, we investigated the expression levels of seven members of the *HbSCAMP* gene family in the stems and bark of rubber trees (Figure 7). In a previous study, we performed transcriptome sequence analysis on the cultivated rubber tree cultivar CATAS73397 and its EMS-induced dwarf mutant MU73397. The raw sequence data are available in the NCBI database under accession number PRJNA1112341. By comparing FPKM values, we found that *HbSCAMP3* was significantly downregulated in the xylem and bark of MU73397 compared to the wild-type, suggesting its potential involvement in stem growth and development. However, the expression of other *HbSCAMP* genes did not show significant changes.

In this study, transgenic poplar lines overexpressing *HbSCAMP3* were constructed to elucidate the potential function of *HbSCAMP3* in regulating plant height. Three independent overexpression lines of *HbSCAMP3* (OE1, OE2, OE3) were selected for subsequent validation. Compared to the wild-type poplar, the HbSCAMP3-OE lines showed significant phenotypic changes. The height of the HbSCAMP3-OE lines increased about 1-fold (Figure 8a,b). qRT-PCR confirmed that the transcript level of *HbSCAMP3* was significantly increased, about 5-fold, in the transgenic lines (Figure 8c). This result is consistent with the phenotypic observations, further supporting the important role of *HbSCAMP3* in the regulation of plant growth. We also conducted a subcellular localization study in tobacco cells. The results showed that the *HbSCAMP3* protein was localized to the plasma membrane (Figure 8d), which is consistent with our previous prediction (Appendix A). The plasma membrane localization suggests that *HbSCAMP3* may be involved in the transport of substances across the cell membrane, which may be related to the mechanism of plant growth.

## 3. Discussion

Understanding the mechanisms of rubber tree height growth is of significant importance for refining breeding programs and optimizing production, as plant height is a key factor affecting plant structure, biomass, and ultimately yield [16,17]. As an important industrial raw material crop, research into the genetic regulatory mechanisms of rubber tree height holds significant practical value for variety improvement and is also crucial for understanding the fundamental processes of plant growth and development. However, the molecular mechanisms underlying the height growth of rubber trees are still poorly understood, particularly the role of the *SCAMP* gene family. *SCAMP* genes are involved in cellular processes such as membrane transport, exocytosis, and endocytosis in plants [18], but their role in height growth warrants further investigation.

### 3.1. SCAMP Gene Family Analysis in Rubber Trees

In this study, we performed a detailed analysis of the *SCAMP* gene family in the rubber tree genome and identified seven members. These HbSCAMP proteins exhibit significant differences in amino acid length and molecular weight, which may indicate the diverse roles they play in the biological processes of the rubber tree. Compared to the differences in amino acid lengths and molecular weights of the *SCAMP* family in *A. thaliana*, the variations among rubber tree SCAMP proteins are even more pronounced [19], potentially reflecting the need in the rubber tree for functional differentiation among members of the *SCAMP* gene family during adaptation to its unique ecological niche. Through the construction of a phylogenetic tree, we found that *HbSCAMPs* have formed three distinct subfamilies compared to *SCAMP* genes from other species, which differs from the situation in soybean where *SCAMPs* are divided into two subfamilies [13]. This diversity and balanced distribution of subfamilies may indicate the adaptation and differentiation of the rubber tree *SCAMP* gene family to various biological functions during the evolutionary process. Further analysis of gene structure and conserved motifs revealed patterns related to phylogenetic relationships, indicating differences in gene structure and conserved motifs between different subfamilies, while members of the same subfamily are essentially the same. This conservation of structure and motifs may be closely related to their specific biological functions in the rubber tree, such as cell signaling or membrane transport processes.

The analysis of CREs in the promoter regions of the *HbSCAMP* gene family shows that these genes are mainly associated with hormone response elements, growth and development elements, and stress response elements. This is similar to the research in soybeans, where *GmSCAMPs* may respond to various abiotic stresses, including hormones, drought, and cold [13]. The presence of these CREs suggests that *HbSCAMPs* may play a key role in the response of rubber trees to environmental changes, especially in regulating plant hormones and coping with environmental stress. The analysis of gene duplication events indicates that there were four pairs of gene duplication events in the *HbSCAMPs*, which is an important mechanism for generating adaptive diversity in the evolution of eukaryotic genomes [20]. The Ka/Ks ratios of these gene pairs are all less than 1, indicating that they are under the constraint of purifying selection during the evolutionary process to maintain functional integrity [21].

### 3.2. Regulation of Plant Stress Response by HbSCAMPs

The *HbSCAMP* gene family encodes secretory carrier membrane proteins that regulate ion movement through cellular channels, promote xylem loading processes, and facilitate the transfer of sugar molecules from photosynthetic tissues to other parts of the plant through phloem loading [22]. Efficient utilization of membrane transporters can enhance tolerance to various abiotic stresses [23].

The high expression of *HbSCAMP1-6* in petioles may be related to their role in substance transport within these tissues. Membrane transporters in petioles increase the supply of water and sugars to all plant organs under abiotic stresses such as drought, cold, and high temperatures, thereby supporting plant growth and development under adverse conditions [24]. For example, the aquaporin *NtAQP1*, which is highly expressed in petioles, is involved in the efficient transport of water, urea, and glycerol [25]. The high expression of *HbSCAMP7* in the stem apex may be related to its role in protecting these regions against stress. Stem tips are the most active growing regions in plants. For example, overexpression of the membrane transporter *OsLCT2* limits cadmium transfer from roots to stems, reducing cadmium accumulation in rice stems and grains, thereby protecting stem tips from toxic metal accumulation and alleviating stress [26].

The transporter MsNTF2L in alfalfa enhances drought resistance by inducing stomatal closure and reducing leaf water loss [27]. In this study, *HbSCAMP1*, *HbSCAMP2*, *HbSCAMP3*, *HbSCAMP6*, and *HbSCAMP7* were significantly upregulated in leaves under drought stress, suggesting that these genes may be involved in similar drought resistance mechanisms. Reduced expression of membrane transporters under salt stress can lead to an imbalance in Na^+^/K^+^ homeostasis, affecting salt tolerance in plants [28]. Under salt stress conditions, *HbSCAMP7* was significantly upregulated in leaves, petioles, and stems. Mechanical damage can trigger stress responses similar to those caused by abiotic stresses such as drought. In these cases, membrane transporters promote the effective transport of key molecules and maintain ion balance, thereby enhancing the plant’s ability to cope with physical damage and recover from it, thus contributing to overall stress tolerance and survival [24,29]. The upregulated expression of *HbSCAMP1*, *HbSCAMP2*, *HbSCAMP5*, and *HbSCAMP7* in bark under wound stress conditions suggests that these genes may be involved in wound healing and defense responses.

### 3.3. Regulation of Plant Growth by HbSCAMP3

In this study, we explored the gene function of *HbSCAMP3* within the *SCAMP* gene family of the rubber tree. The higher expression of *SCAMPs* in the xylem is associated with increased wood density and the development of secondary walls, which are crucial for the structural integrity of the plant and water transport [15]. In the CATAS73397 rubber tree, the expression of the *HbSCAMP3* gene in the xylem is higher than in the bark, which may relate to its role in the transport of substances within the plant. The synthesis of cell walls and secondary growth in the xylem requires more material transport and accumulation of cell wall precursors compared to the bark [30]. In the dwarf mutant MU73397 of the rubber tree, the expression level of HbSCAMP3 is significantly reduced, leading to a decreased supply of materials necessary for cell wall synthesis and secondary growth, resulting in slower plant growth.

Furthermore, the plasma membrane localization of HbSCAMP3 in tobacco leaf epidermal cells indicates its potential involvement in intercellular transport. In tobacco, *SCAMP2* is known as a marker for plant cell secretory vesicles, facilitating the transport of secretory materials from the Golgi apparatus to the plasma membrane and cell plate, forming clusters of secretory vesicles [18]. Similarly, *SCAMP1*, which has four transmembrane domains, reaches the plasma membrane through the ER–Golgi–TGN–PM pathway in plant cells, highlighting the importance of its N-terminus and transmembrane domains in the transport process [31]. Therefore, we speculate that *HbSCAMP3* may participate in the plant growth process by transporting important substances, such as extracellular polysaccharides, through extracellular and intracellular membrane transport pathways to the growth sites in plant cells.

## 4. Materials and Methods

### 4.1. Identification and Sequence Analysis of SCAMP Genes of Rubber Trees

The rubber tree genome was accessed in NCBI and used to develop an HMM model using SCAMP protein sequences from different species. This model was then used to filter the rubber tree protein database, identifying SCAMP-like sequences with an e-value limit of 1 × 10^−5^. Duplicates were removed, yielding a set of potential SCAMP sequences. These were confirmed using CDD and SMART with the same e-value threshold, ensuring the presence of the SCAMP domain. The confirmed genes were named based on their genomic locations. Their subcellular localization was predicted with Cell-PLoc 2.0, and protein characteristics such as molecular weight, isoelectric point, instability index, and hydropathy were analyzed using ProtParam from ExPASy (https://web.expasy.org/protparam/ accessed on 15 April 2024).

### 4.2. Phylogenetic Relationship, Gene Structure, and Conserved Motif and Syntenic Analysis of SCAMPs

Alignments of SCAMP proteins from *Hevea brasiliensis*, *Manihot esculenta*, *Populus trichocarpa*, and *Arabidopsis thaliana* were executed with ClustalW (https://www.genome.jp/tools-bin/clustalw/ accessed on 15 April 2024). An ML phylogenetic tree was constructed using IQ-TREE 2 software, with 1000 bootstrap replicates for validation. Data on chromosomal positions and exon–intron structures of *SCAMPs* were obtained from Ensembl Plants and visualized using MapGene2Chromosome (https://qiaoyundeng.github.io/#:~:text=input1:%20Ge/ accessed on 15 April 2024). GSDS 2.0 was used to illustrate the exon–intron structures, while MEME identified conserved motifs within the sequences. The promoter region of *SCAMPs* was examined for regulatory elements with PlantCare. The reference genome of the rubber tree (MT/VB/25A 57/8) was sourced from NCBI [32]. Collinearity of *SCAMP* genes were analyzed with the MCScanX tool in TBtools (version 2024.1.11). Selective pressures on *HbSCAMPs* were assessed using Ka and Ks values, calculated by MCScanX.

### 4.3. Subcellular Localization of HbSCAMP3

The subcellular distribution of the *HbSCAMP3* protein was determined via a transient expression assay in Nicotiana benthamiana leaf cells. A recombinant construct was engineered, integrating the *HbSCAMP3* gene with EGFP within the pBI121 vector, for the purpose of the assay. A parallel control was set up using the pBI121 vector with EGFP alone. Nuclear staining was performed using DAPI, while FM4-64 was employed to outline the cell membrane. The Agrobacterium tumefaciens strain carrying the pBI121-HbSCAMP3-EGFP plasmid was incubated at 28 °C for an appropriate period prior to infiltration into the tobacco leaf epidermis. After transformation, the plants were kept in a controlled environment for 48 h to facilitate protein expression. The subcellular localization was captured using a confocal laser scanning microscope (TCS-SP8MP, Leica, Wetzlar, Germany).

### 4.4. Construction and Transformation of HbSCAMP3 Overexpression Vector in Poplar

Total RNA was first extracted from rubber tree leaves using the RNAprep Pure Polyphenol Plant Total RNA Extraction Kit. Subsequently, cDNA was synthesized using the StarScript Pro All-in-one RT Mix with gDNA Remover for reverse transcription. Specific primers containing *Xho* I and *Avr* II restriction enzyme sites were then designed to amplify the target gene *HbSCAMP3* via PCR. Following this, the digested *HbSCAMP3* gene fragment was ligated with the similarly digested pEarleyGate 100 overexpression vector, resulting in the construction of the pEarleyGate 100 overexpression vector containing *HbSCAMP3*. This constructed vector was then transformed into *Agrobacterium* GV3101, which facilitated the genetic transformation of wild-type poplar 895. Positive transformants were identified through antibiotic resistance screening to confirm stable transgene integration. The genetic transformation of the poplar was carried out by Shanghai Waker Biotechnology Co., Ltd. (Shanghai, China) The primers used in this study are listed in Appendix A. Quantitative real-time PCR was conducted using the 2^−ΔΔCT^ method to assess gene expression levels, and three plant lines exhibiting a significant upregulation in gene expression were chosen for subsequent functional analysis.

### 4.5. Plant Stress Treatment and qRT-PCR Analysis

To study the expression patterns of *HbSCAMP* gene family members under different stress conditions, we used qRT-PCR to validate their expression in different tissues of rubber trees. One-year-old healthy budded rubber tree seedlings were used as experimental material. Stress treatments were as follows: for drought stress, plants were irrigated with 300 mM mannitol solution for 7 days; for salt stress, plants were irrigated with 200 mM NaCl solution for 7 days; for wound stress, uniform incisions (1 cm) were made on the bark 10 cm above the ground level using sharp blades, with one incision per day for 7 consecutive days. Each stress condition included three biological replicates. Sampling sites were stems, petioles, leaves, stem apex, and bark. Samples were immediately frozen in liquid nitrogen and stored at −80 °C prior to qRT-PCR analysis. All stress treatments lasted for 7 days.

For the qRT-PCR procedure, samples from multiple tissues were collected. RNA was isolated using the RNAprep Pure Polyphenol Plant Total RNA Extraction Kit, and cDNA was synthesized using the StarScript Pro All-in-one RT Mix with gDNA Remover, following the manufacturer’s protocol (GeneStar, Beijing, China). Primers specific for the gene of interest and the internal control HbActin are detailed in Appendix A. The qRT-PCR was performed using the SYBR PreMix Ex Taq II (TaKaRa, Otsu, Japan), in accordance with the manufacturer’s recommendations.

## 5. Conclusions

In this study, we identified and characterized seven *HbSCAMP* genes in the rubber tree genome, revealing their evolutionary relationships and genomic structures. The high expression of *HbSCAMP1-6* in petioles and *HbSCAMP7* in stem apex, along with their distinct responses to drought, salt, and wound stresses, suggests their critical roles in substance transport and stress adaptation. The overexpression of *HbSCAMP3* in poplar demonstrated its role in plant growth regulation. These findings indicate the crucial role of the *SCAMP* gene family in the rubber tree’s adaptation to adversity, particularly highlighting that *HbSCAMP3* can promote plant growth. This provides new candidate genes for enhancing the rubber tree’s stress adaptability and improving the genetic traits related to plant height.

## Figures and Tables

**Figure 1 plants-13-02729-f001:**
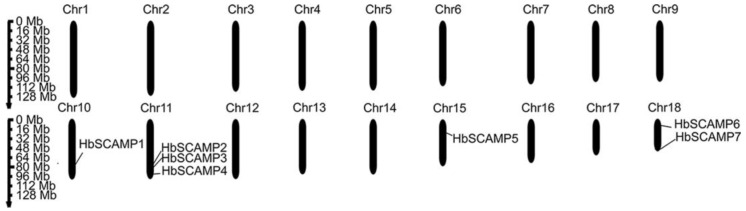
Chromosomal locations of the HbSCAMP genes on the eighteen rubber tree chromosomes.

**Figure 2 plants-13-02729-f002:**
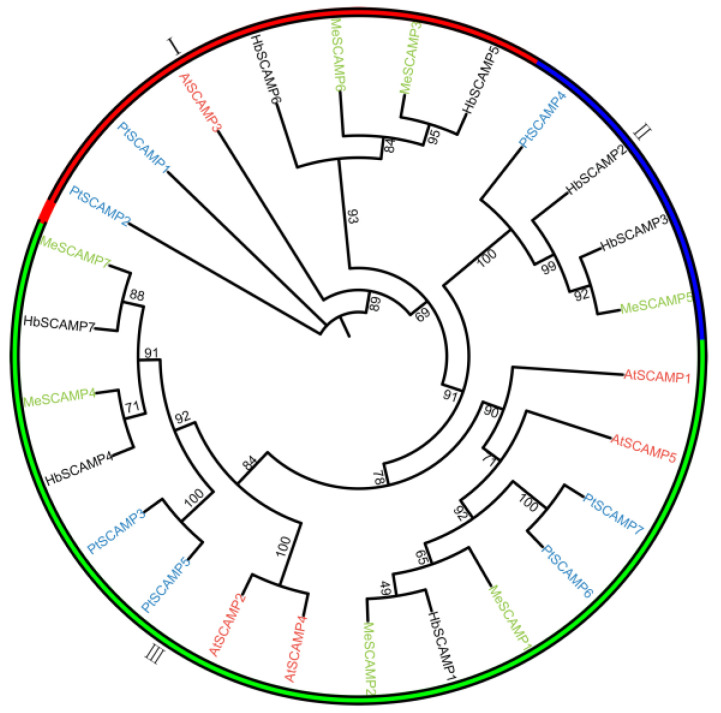
Phylogenetic analysis of SCAMP proteins across *H. brasiliensis*, *A. thaliana*, *M. esculenta*, and *P. trichocarpa*. A maximum likelihood (ML) phylogenetic tree was constructed using IQ-TREE (version 2.0) software with 1000 bootstrap replicates to assess branch support. The distinct colors of the outer ring and branches correspond to the three identified subfamilies of the SCAMP gene family. Ⅰ, subfamily Ⅰ; Ⅱ, subfamily Ⅱ; Ⅲ, subfamily Ⅲ.

**Figure 3 plants-13-02729-f003:**
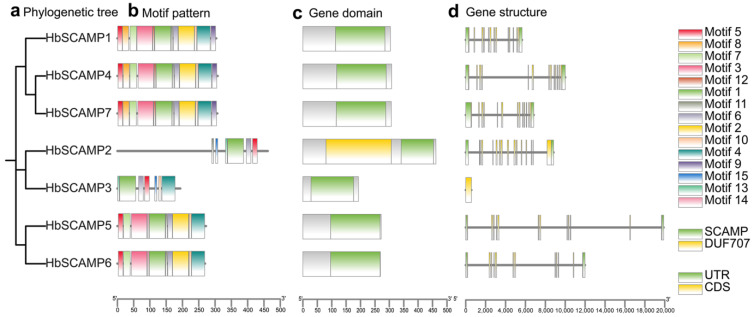
Comprehensive analysis of HbSCAMPs, including phylogenetic relationships, conserved motifs, and gene structures. (**a**) The maximum likelihood (ML) phylogenetic tree of HbSCAMP proteins was constructed from full-length sequences using IQ-TREE software with 1000 bootstrap replicates to estimate branch support. Branch colors indicate different subfamilies. (**b**) The distribution of conserved motifs in HbSCAMP proteins, predicted using the MEME suite. A total of 15 motifs are displayed, each represented by a unique color. The scale bar represents 50 amino acids (aa). (**c**) Schematic representation of the SCAMP domain distribution across HbSCAMP proteins, highlighting the presence of the domain in all family members. (**d**) Gene structures of HbSCAMPs, depicted with introns (black lines), exons (colored rectangles), and untranslated regions (UTRs, gray rectangles). Exon colors correspond to the domains they encode. The scale bar indicates 2000 bp.

**Figure 4 plants-13-02729-f004:**
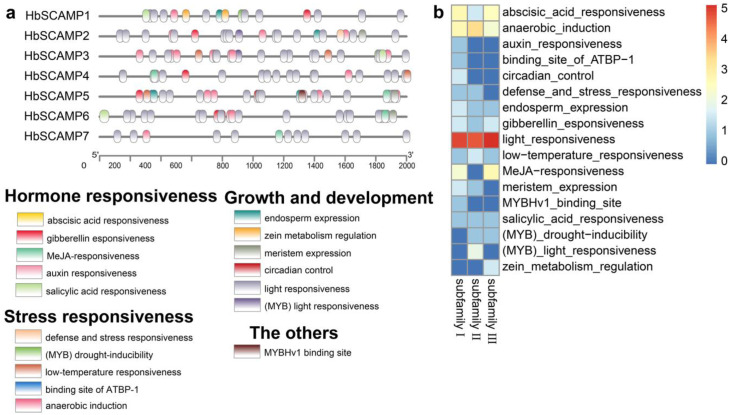
Analysis of cis-regulatory elements (CREs) within the putative promoter regions of HbSCAMP genes. (**a**) Distribution of CREs within the 2000 bp upstream promoter region of HbSCAMP genes. (**b**) Quantitative representation of CREs in the promoters of HbSCAMP genes depicted as a heatmap. To enhance the visualization of variations in CRE density, the values presented in the heatmap are the log2 transformation of the actual CRE counts.

**Figure 5 plants-13-02729-f005:**
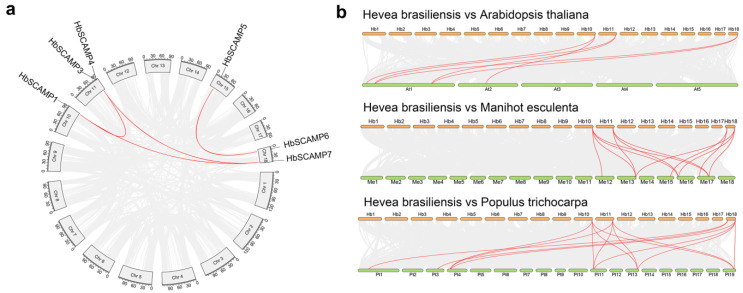
Synteny analysis of SCAMP genes in *Hevea brasiliensis*, *Arabidopsis thaliana*, *Manihot esculenta*, and *Populus trichocarpa*. (**a**) Chromosome map of *H. brasiliensis* (Hb), displaying its 18 chromosomes (Chr 1–18). The chromosome length is measured in megabases (Mb). Homologous *SCAMP* gene pairs are indicated by red lines. (**b**) Comparative chromosome maps of *H. brasiliensis* (Hb, Hb1–8), *A. thaliana* (At, At1–5), *M. esculenta* (Me, Me1–18), and *P. trichocarpa* (Pt, Pt1–19) are presented. Red lines denote homologous *SCAMP* gene pairs.

**Figure 6 plants-13-02729-f006:**
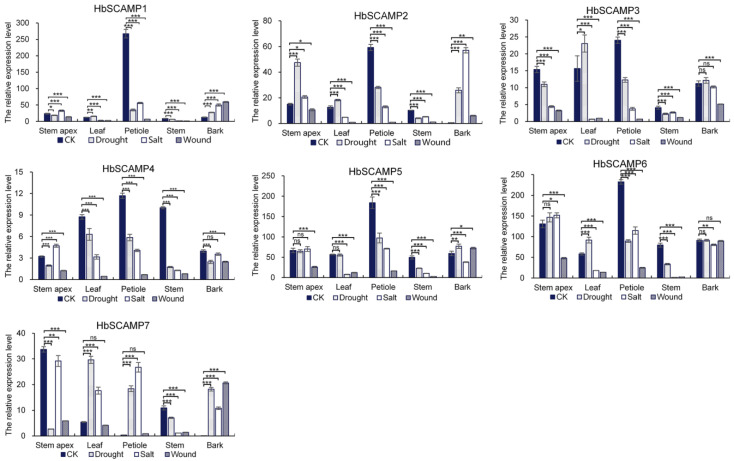
qRT-PCR analysis of HbSCAMP expression in rubber trees under different stress treatments. The expression levels of HbSCAMP genes in various tissues were evaluated under drought (300 mM mannitol), salt (200 mM NaCl), and wound (7 stem incisions) stress conditions. All experiments were performed independently at least three times. Error bars represent the standard deviation of the three replicates. Asterisks indicate significant differences in transcript levels compared to the control (“ns” indicates not significant, * *p* < 0.05, ** *p* < 0.01, *** *p* < 0.001).

**Figure 7 plants-13-02729-f007:**
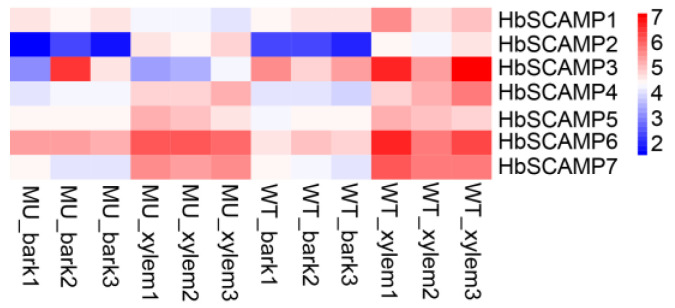
HbSCAMP expression in rubber tree xylem. The color scale from blue to red represents log2-transformed FPKM values ranging from low to high. “MU” represents the dwarf line MU73397 and “WT” represents the wild-type control CATAS73397.

**Figure 8 plants-13-02729-f008:**
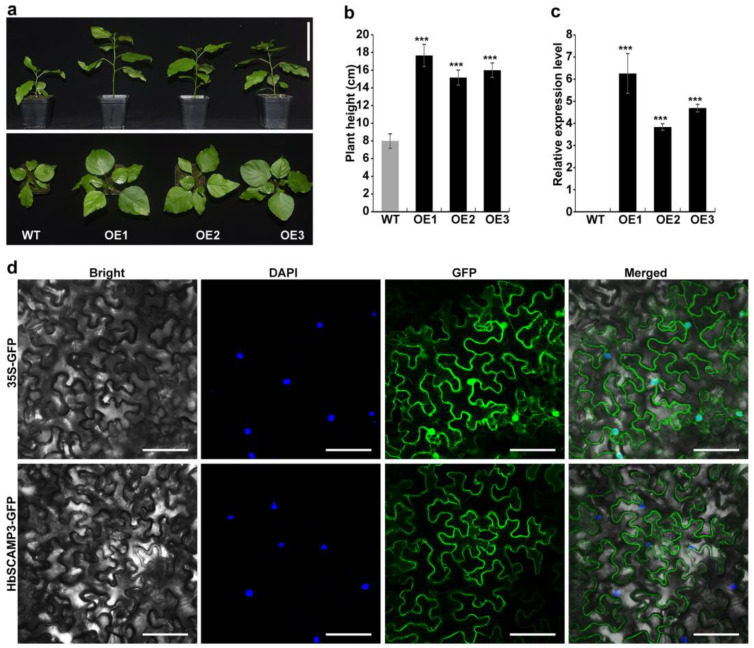
The gene function of HbSCAMP3. (**a**) Phenotypic picture of HbSCAMP3-OE lines (OE1, OE2, and OE3) and WT poplar. Bar = 10 cm. (**b**) The height statistics of HbSCAMP3-OE lines (OE1, OE2, and OE3) and WT poplar. (**c**) The expression level of HbSCAMP3 in poplar. *** *p* < 0.01. “WT” represents the wild-type, while “OE1, OE2, and OE3” refer to three lines overexpressing HbSCAMP3 in poplar. PtActin was used as an internal control for calculating the relative expression levels of the above genes. Values are shown as mean ± SD (n = 3). (**d**) Subcellular localization of HbSCAMP3 in tobacco leaf epidermal cells. The blue dots represent DAPI staining of the cell nuclei. The green color represents green fluorescent protein (GFP). Bar = 25 µm.

## Data Availability

Data are contained within the article and the Appendix A.

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
