# Peer review of "Genome-Wide Identification of Rubber Tree SCAMP Genes and Functional Characterization of HbSCAMP3"

_plants, 2024, doi:10.3390/plants13192729_

Round 1

Reviewer 1 Report

Comments and Suggestions for Authors

PLEASE CHECK GRAMMAR AND SPELLING MISTAKES THROUGHOUT THE PAPER.

GIVE AIM OF THE STUDY IN UNDERSTANDABLE APPROPRIATE WAY.

THE findings highlight the potential of 419 HbSCAMP genes in enhancing plant stress tolerance and underscore the specific role of 420 HbSCAMP3 in plant growth regulation, providing a foundation for future genetic improvement of rubber tree.

Although the paper is a comprehensive work carried out by the authors on the subject, many sentences need rewrting e.g.

As an important 252 industrial raw material crop, research into the genetic regulatory mechanisms of rubber 253 tree height not only holds significant practical value for variety improvement but is also 254 crucial for understanding the fundamental processes of plant growth and development.

Could be written as

As an important industrial raw material crop, research into the genetic regulatory mechanisms of rubber tree height holds significant practical value for variety improvement and is also crucial for understanding the fundamental processes of plant growth and development.

If these sentences are rewritten, the paper should be accepted.

Comments on the Quality of English Language

minor revision is desired in the language and construction of the sentences.

Author Response

Comments 1: PLEASE CHECK GRAMMAR AND SPELLING MISTAKES THROUGHOUT THE PAPER.

Response 1: Thank you for pointing this out. We have thoroughly reviewed the manuscript for grammar and spelling errors and made the necessary corrections. The revised text can be found in red in the manuscript.

Comments 2: GIVE AIM OF THE STUDY IN UNDERSTANDABLE APPROPRIATE WAY. THE findings highlight the potential of 419 HbSCAMP genes in enhancing plant stress tolerance and underscore the specific role of 420 HbSCAMP3 in plant growth regulation, providing a foundation for future genetic improvement of rubber tree.

Response 2: Thank you for pointing this out. We have revised the text from lines 420 to 423 of the manuscript. Below are the modified sentences. “These findings indicate the crucial role of the SCAMP gene family in the rubber tree's adaptation to adversity, particularly highlighting that HbSCAMP3 can promote plant growth. This provides new candidate genes for enhancing the rubber tree's stress adaptability and improving the genetic traits related to plant height.” The revised text can be found in red in the manuscript.

Comments 3: Although the paper is a comprehensive work carried out by the authors on the subject, many sentences need rewrting e.g. As an important 252 industrial raw material crop, research into the genetic regulatory mechanisms of rubber 253 tree height not only holds significant practical value for variety improvement but is also 254 crucial for understanding the fundamental processes of plant growth and development. Could be written as “As an important industrial raw material crop, research into the genetic regulatory mechanisms of rubber tree height holds significant practical value for variety improvement and is also crucial for understanding the fundamental processes of plant growth and development”.

Response 3: Thank you for your feedback. We have made revisions to lines 250 to 253 of the manuscript based on your suggestions and highlighted the changes in red.

Comments 4: Minor revision is desired in the language and construction of the sentences.

Response 4: Thank you for your suggestion. We have made revisions to improve the language and sentence construction throughout the manuscript. All changes have been highlighted in red.

Reviewer 2 Report

Comments and Suggestions for Authors

The work of Yang et al. offer insights about the presence and regulation of SCAMP coding  genes on Hevea brasilensis. The bioinformatic characterization and expression analysis offer useful information for readers. The more interesting experiment is the generation of SCAMP3-overexpression Poplar plants, but only their size measure is reported. In this regard, authors should make a better description of the protocol used to generate these plants and cite the corresponding work related to the generation of transgenic plants. Otherwise, authors must report a complete protocol, in order to their findings can be reproduced by others.

Author Response

Comments 1: The work of Yang et al. offer insights about the presence and regulation of SCAMP coding genes on Hevea brasilensis. The bioinformatic characterization and expression analysis offer useful information for readers. The more interesting experiment is the generation of SCAMP3-overexpression Poplar plants, but only their size measure is reported. In this regard, authors should make a better description of the protocol used to generate these plants and cite the corresponding work related to the generation of transgenic plants. Otherwise, authors must report a complete protocol, in order to their findings can be reproduced by others.

Response 1: Thank you for your valuable feedback. We appreciate your suggestions regarding the description of the protocol for generating SCAMP3-overexpression poplar plants. We have included a more detailed protocol in the revised manuscript on the generation of transgenic plants. This information can be found in the revised manuscript on lines 380 to 394. We hope this enhances the reproducibility of our findings.

Reviewer 3 Report

Comments and Suggestions for Authors

The whole-genome analysis of rubber plants to identify the SCAMP gene family members and their response to multiple abiotic stress by Yang et al., is very interesting. Here author performed a comprehensive molecular study to defend the hypothesis. Moreover, author also explored that SCAMP gene family members involved in regulation of plant growth that provide new insights of how plant acclimated to environmental variability. The structure of the MS is perfect and could be acceptable, however some advice are as follows

-          Many of the figures not clear author suggested to provide clear images.

-          - Author suggest to provide logical expiation in discussion part why the SCAMP genes differential expressed in bark or xylem of rubber plants.

Author Response

Comments 1: Many of the figures not clear author suggested to provide clear images.

Response 1: Thank you for your valuable feedback. We have improved the clarity of the figures by providing higher resolution images. The revised figures are now clearer and easier to interpret.

Comments 2: Author suggest to provide logical expiation in discussion part why the SCAMP genes differential expressed in bark or xylem of rubber plants.

Response 2: Thank you for pointing this out. We have added an explanation regarding the differential expression of the SCAMP gene in the bark and xylem of rubber trees in lines 322 to 332 of the manuscript. The specific addition is as follows:

 “In this study, we explored the gene function of HbSCAMP3 within the SCAMP gene family of the rubber tree. The higher expression of SCAMPs in the xylem is associated with increased wood density and the development of secondary walls, which are crucial for the structural integrity of the plant and water transport [15]. In the CATAS73397 rubber tree, the expression of the HbSCAMP3 gene in the xylem is higher than in the bark, which may relate to its role in the transport of substances within the plant. The synthesis of cell walls and secondary growth in the xylem requires more material transport and accumulation of cell wall precursors compared to the bark [31]. In the dwarf mutant MU73397 of the rubber tree, the expression level of HbSCAMP3 is significantly reduced, leading to a decreased supply of materials necessary for cell wall synthesis and secondary growth, resulting in slower plant growth.”